# How Did People with Prediabetes Who Attended the Diabetes Prevention Education Program (DiPEP) Experience Making Lifestyle Changes? A Qualitative Study in Nepal

**DOI:** 10.3390/ijerph20065054

**Published:** 2023-03-13

**Authors:** Pushpanjali Shakya, Monish Bajracharya, Eva Skovlund, Abha Shrestha, Biraj Man Karmacharya, Bård Eirik Kulseng, Abhijit Sen, Aslak Steinsbekk, Archana Shrestha

**Affiliations:** 1Department of Public Health and Nursing, Norwegian University of Science and Technology, 7491 Trondheim, Norway; 2Department of Business and IT, University of South-Eastern Norway, 3800 Bø, Norway; 3Department of Community Medicine, Kathmandu University School of Medical Sciences (KUSMS), Dhulikhel 45200, Nepal; 4Department of Public Health and Community Programs, Kathmandu University School of Medical Sciences (KUSMS), Dhulikhel 45200, Nepal; 5Department of Clinical and Molecular Medicine, Norwegian University of Science and Technology, 7491 Trondheim, Norway; 6Centre for Oral Health Services and Research (TkMidt), 7030 Trondheim, Norway; 7Institute for Implementation Science and Health, Kathmandu 44600, Nepal; 8Department of Chronic Disease Epidemiology, Yale School of Public Health, New Haven, CT 06520-0834, USA

**Keywords:** benefits, hurdles, lifestyle changes, prediabetes, understanding

## Abstract

Diabetes can be prevented through lifestyle modification in the prediabetic phase. A group-based lifestyle intervention called ‘Diabetes Prevention Education Program’ (DiPEP) was tested recently in Nepal. The present study aimed to explore experiences of making lifestyle changes among people with prediabetes participating in the DiPEP. This qualitative study, with semi-structured interviews of 20 participants, was conducted 4–7 months following DiPEP intervention. Data analysis was performed by thematic analysis. The results included four themes, understanding that diabetes could be prevented, lifestyle changes made, hurdles to overcome, and experiencing benefits leading to sustained change. Some participants said they felt relieved to know that they had a chance to prevent diabetes. The participants talked mostly about making changes in diet (reducing carbohydrate intake) and physical activity (starting exercises). Obstacles mentioned included a lack of motivation and a lack of family support to implement changes. Experiencing benefits such as weight loss and reduced blood sugar levels were reported to lead them to maintain the changes they had made. Understanding that diabetes could be prevented was a key motivator for implementing changes. The benefits and hurdles experienced by the participants of the present study can be taken into consideration while designing lifestyle intervention programs in similar settings.

## 1. Introduction

Prediabetes is a health condition with higher than normal blood glucose level [fasting plasma glucose 100–125 mg/dL or 2-h plasma glucose 140–199 mg/dL or glycated hemoglobin (HbA1c) 5.7–6.4%], but not yet meeting the threshold of diabetes diagnosis [1]. The prevalence of prediabetes is increasing worldwide, with the highest age-adjusted prevalence in low-income countries [2]. In Nepal, the prevalence of Impaired Fasting Glucose (IFG) and Impaired Glucose Tolerance (IGT) were found to be 7.8% and 5.4%, respectively, as per the International Diabetes Federation Atlas [2], while other previous studies showed the prevalence of prediabetes determined by IFG and IGT ranged from 1.3–19.4% [3,4,5,6,7]. A meta-analysis has shown that the prevalence of prediabetes in Nepal was 9.2% in 2020 [8]. In the urban settings of Nepal, the prevalence of prediabetes determined by HbA1c was 5% [9]. The annual risk of development of type 2 diabetes (T2D) among individuals with prediabetes has been estimated to be 5–10% [10,11]. One of the strongest determinants of prediabetes is obesity [12], which is also associated with metabolic syndrome [13]. The associated comorbidities of prediabetes increase health care expenditure and deteriorate the quality of life [14] as well as increase mortalities [15,16].

T2D is one of the fastest-growing global health emergencies of the 21st century, causing an estimated 6.7 million deaths in 2021 [2]. The prevalence of T2D was 10.3% in Nepal as per meta-analysis and systematic review [17]. In the context of developing countries with out-of-pocket healthcare expenditure systems, such as Nepal [18], the economic burden associated with the disease is carried by the individuals and their families [19,20]. Considering all the drawbacks of T2D, it is urgent to prevent or delay it. Prevention or delay of T2D is possible with non-pharmacological interventions, including a healthy diet and adequate physical activity targeting weight reduction during the prediabetes phase [21]. One such evidence-based intervention is the Diabetes Prevention Program (DPP) developed in the US, consisting of 16 sessions mostly conducted in clinical settings with follow-up by lifestyle coaches [22]. This program can lead to a 30–60% reduction in the incidence of T2D among adults with high risk [23] and, in some cases, conversion back to normoglycemia [24]. However, most of the studies included in a systematic review of lifestyle intervention programs were undertaken in developed nations [11]. There is a dearth of evidence on the effectiveness of community-level interventions in the context of Nepal. Though some national programs deliver diabetes management interventions, the prevention of diabetes has not been prioritized in Nepal [25,26].

Globally, qualitative studies have investigated perspectives on prediabetes education intervention among different stakeholders, including user groups [27,28,29]. There are also qualitative studies exploring perspectives of individuals with prediabetes making dietary changes and physical activity changes following the diagnosis of prediabetes and participation in lifestyle intervention programs [30,31,32,33,34,35]. These studies have revealed facilitators and barriers for lifestyle changes among people with prediabetes in different parts of the world [30,35]. However, such studies have mostly been conducted in high-income countries.

Exploring the experiences and perceptions of the stakeholders, especially the users group, is substantial to emphasize the effectiveness of the interventions and also is important in improving the content and form of interventions. Such studies are scarce in countries with low-resource settings. To the researchers’ knowledge, no studies in Nepal have yet explored or reported experiences of people with prediabetes who underwent diabetes prevention intervention programs at the community level; such experiences can be helpful to reach people better and/or improve certain aspects of the interventions. Recently, a community-based cluster randomized controlled trial (RCT) investigating the effect of an intervention called “Diabetes Prevention Education Program (DiPEP)” was conducted in Nepal [36]. It was a group-based lifestyle intervention program designed with inspiration from the National Diabetes Education Program [37] and the DPP [22] from the USA. The aim of the DiPEP intervention was to prevent diabetes among people with prediabetes in the community settings of Nepal. The intervention targeted people with prediabetes and consisted of four one-hour weekly educational sessions and follow-ups for the next five months. The trial provided an opportunity to explore participants’ experiences of making lifestyle changes.

Hence, the aim of the present qualitative study was to explore the experiences of making lifestyle changes among individuals with prediabetes who participated in the DiPEP intervention.

## 2. Materials and Methods

### 2.1. Study Design

This was a qualitative study following a phenomenological approach with semi-structured interviews conducted at the individual-level from August 2020 to January 2021 as part of a larger project, including screening campaigns in the community to screen for individuals with prediabetes and a cluster RCT on the effect of the DiPEP intervention. The study protocol for the larger project is published elsewhere [36]. The consolidated criteria for reporting qualitative studies (COREQ): a 32-item checklist [38] was consulted to report this study.

### 2.2. Setting and Intervention

The RCT took place in two urban community settings in Nepal (Patan and Dhulikhel) [9,36]. However, due to the practical consequences of the COVID-19 lockdown, the qualitative study included participants in the intervention arm of the RCT from Patan only. Patan is a core part of Lalitpur Metropolitan City located 5 km southeast of the capital Kathmandu, densely populated with a population of 284,922, a literacy rate of 80%, and an increasing trend of westernization [39].

As part of the larger project, screening campaigns were organized in the communities among the general public with the eligibility criteria: (i) a permanent resident of the study sites, (ii) age 18–64 years, and (iii) no self-reported history of diabetes [36]. Banners and verbal announcements using speakers about the screening campaigns in the Nepali language were used to let the general public know about the campaigns. The screening campaigns for the RCT were aimed at detecting persons with prediabetes, defined by HbA1c ranging from 5.7–6.4% [36]. Those identified as having prediabetes were informed that their blood sugar level was higher than normal, but they had not yet reached the threshold of diabetes. They were also provided with information about the RCT.

The DiPEP is a 6-month, group-based intervention program in the community setting comprising four one-hour (30 min theory + 30 min practical) weekly educational sessions held during the first month of the intervention and follow-up for the next five months. The educational sessions were conducted physically in Patan prior to the COVID-19 pandemic and digitally during the lockdown phase. The topics of the four educational sessions were: (i) introduction to diabetes and prediabetes, (ii) healthy eating and physical activity, (iii) stress management, and (iv) management of social cues. The local and cultural context of Nepali users was considered during the development of the intervention, and the intervention was delivered in the Nepali language. Examples of considering local and cultural context included an emphasis on adjusting the sizes of portions of available food instead of asking the participants to introduce new types of food under the topic of ‘healthy diet’ and including different types of simple exercises that could be performed at home under the topic of ‘physical activity’. Five goals of the intervention were clearly conveyed at the beginning of the educational sessions. The goals were: (1) to prevent diabetes, (2) to reduce body weight by 5–7%, (3) to perform at least 150 min of moderate exercise per week, (4) to gain support from family and friends, and (5) to overcome the obstacles during lifestyle changes. The researcher (PS) and study nurses conducted the educational sessions. Participants were provided with written materials consisting of a diabetes prevention education brochure (including a summary of the four topics of DiPEP educational sessions), an exercise calendar, and a food logbook [36].

The five months of follow-up of the participants prior to the pandemic included group-based weekly physical sessions conducted by community health care workers/volunteers (CHCW/Vs) and group-based monthly physical meetings with both the CHCW/Vs and the study nurses. The physical follow-up included measurement of weight and blood pressure, assessment of the food logbook and exercise calendar, doing four types of exercises, and a question–answer (Q&A) session. During the COVID-19 lockdown, these physical follow-ups were replaced with individual biweekly telephone calls by the CHCW/Vs and group-based monthly digital meetings conducted by the researcher (PS). In the first telephone call of the month, CHCW/Vs reminded participants about the DiPEP lessons and answered any questions asked by the participants; in the second telephone call, they also invited participants to the monthly digital meeting. The digital meeting included the revision of four topics, asking about the food logbook and the exercise calendar, doing four types of exercises (by displaying a video of the same exercises), and a Q&A session.

### 2.3. Study Participants

Individuals included in the present qualitative study met the following inclusion criteria: prediabetes detected by HbA1c 5.7–6.4%; age of 18–64 years; participation in the intervention arm of the RCT [9,36]; attendance at a minimum of one of the four educational sessions; access to the internet, as the interviews were performed digitally due to COVID-19 lockdown. To ensure diversity in the study sample, participants were purposively selected considering gender, education level, ethnic group, physical or digital participation in the intervention, and the number of educational sessions attended.

The recruitment was performed by selecting the participants from the top of the attendance lists of the educational sessions. The order of participants on the lists was based either on ascending order of the code number that they had from the screening program or their seat position in the first educational session. Written consent was obtained before enrollment in the RCT and after the participants were informed about the interview and the study. Those participants selected for the interviews from the intervention clusters were contacted by phone or digitally. Once a participant agreed to take part in the interview, an information sheet with detailed information about the interview in the Nepali language was sent digitally, and an appointment for the digital interview was made. Participants were aware of the purpose of the interview and the person (PS) who would conduct the interview since the researcher (PS) had dual roles as an educator and as an interviewer. On the day of the interview, as physical contact was avoided due to the pandemic, the interviewer read the information sheet and consented again to the participants on a digital platform. Verbal consent was then obtained from the participants before the interview commenced.

A total of 6222 participants participated in the screening campaigns, and 308 had prediabetes. Out of these, 291 were enrolled in the RCT, and 159 were recruited in the intervention arm. Only 73 participants attended at least one DiPEP educational session and were thus eligible for the present qualitative study. Among these, 31 participants were approached for the interview; these were selected from the list of participants in the attendance sheet of the educational sessions and also fit the inclusion criteria. Recruitment was stopped when 20 persons had been interviewed, and data saturation had been achieved [40], where no new information relevant to the study was found. The reasons for the non-participation of 11 participants were that they were not reachable (n = 5) or not available for the interview (n = 6).

### 2.4. Data Collection

Data were collected through individual semi-structured face-to-face interviews held in the Nepali language. Due to the restrictions during the pandemic, all interviews were held digitally. For practical reasons, participants who attended digital educational sessions were interviewed four months after the start, while those taking part in physical sessions were interviewed seven months after the start, i.e., three months before and one month after the end of the program, respectively. One researcher (PS, a female Ph.D. scholar trained in qualitative studies at two universities in Norway) conducted all of the interviews. A note keeper made notes of significant verbal or non-verbal actions during the interview. The interviews lasted from 40 to 82 min, with an average of 62 min.

A semi-structured interview guide, based on the literature [41,42,43] thoroughly discussed among the research group and reviewed by the other three external experts, was developed and pre-tested among the study nurses. The main questions were: What was your experience with DiPEP? What were the lifestyle changes implemented afterwards? How easy or difficult was it to follow the DiPEP lessons and maintain those changes? The guide also included several probing questions, and probes were also asked based on the responses of the participants. No changes were made to the main questions during the data collection.

### 2.5. Data Analysis

All interviews were audio-recorded and transcribed verbatim in Nepali by native Nepali speakers (study nurses and research staff). To analyze the data, a thematic cross-case analysis called ‘systematic text condensation’ was used for this paper [44]. This consists of four steps: “(1) total impression—from chaos to themes; (2) identifying and sorting meaning units—from themes to codes; (3) condensation—from code to meaning; (4) synthesizing—from condensation to descriptions and concepts” [44]. This was performed iteratively.

The transcripts were initially read and reviewed by two researchers (PS and MB) against the audio recording for quality assurance and to gain a total impression. First, three transcripts were read independently by PS from a bird’s eye perspective to identify preliminary themes. PS suggested three themes: (i) perception, (ii) knowledge, and (iii) implication. After reading seven other transcripts, PS recategorized and renamed the preliminary themes as: (i) participants’ understanding, (ii) acceptability of having prediabetes, (iii) participants’ perception, (iv) impact of the intervention, and (v) effects of the COVID-19 pandemic. MB read ten transcripts independently and suggested four preliminary themes called (i) understanding, (ii) level of acceptance, (iii) DiPEP and its effectiveness, and (iv) effect of COVID-19. PS and MB identified ‘meaning units’ (the smallest text fragment containing information about the research question) [44] independently and sorted them into their separate preliminary themes. This material was discussed thoroughly by PS and MB. A senior researcher (ArSh) checked the coding and resolved disagreements where necessary. Based on this, for this paper, three main themes called (i) understanding, (ii) acceptance of having prediabetes, and (iii) behavior changes were made. Then PS and MB coded all remaining interviews independently based on these three themes. Sub-themes were identified and defined under the given themes throughout the process of coding.

This was further discussed with two senior researchers (ArSh and AsSt), and the findings were recategorized into four new themes with more focus on the participants’ experience and less on the actual changes they said they had made. The final themes were: (i) understanding that T2D can be prevented, (ii) lifestyle changes made, (iii) hurdles to overcome, and (iv) experiencing benefits leading to sustained change. A new round of sorting of meaning units was then conducted by PS and AsSt, with subsequent development of sub-themes and condensation of these. Finally, the contents of the condensations were rewritten into analytic text [44]. Quotations from each sub-theme that supported the themes were selected from the transcript and translated into English. The quotations were marked with PhyX and DigX, respectively, for participants attending the interventions physically or digitally. Nvivo [version 20.6.1.1137] was used to support the coding process.

## 3. Results

### 3.1. Participants’ Characteristics

The mean age of participants (n = 20) was 51 (SD = 9) years; 50% were female. Fifty percent of the participants had at least a higher secondary level of education and half of the participants (50%) were self-employed (Table 1).

Table 2 presents baseline lifestyle characteristics, anthropometric measurements, and clinical characteristics of the participants. The majority were non-smokers (95%), and half had never consumed alcohol. Physical activity was determined by Global Physical Activity Questionnaire using metabolic equivalent (METs) minutes per week [45]. It included physical activities of various levels, such as mild, moderate, and vigorous activities [45]. More than half (65%) had physical activity (METs) (≥600 min per week), which is recommended by the WHO [45]. The mean total grain intake and mean total fruits/vegetables intake were 637 gm/day (SD = 143) and 264 gm/day (SD = 106), respectively. All had central obesity measured by waist circumference. Mean RBS and HbA1c were 167.6 mg/dL (SD = 33.9) and 5.9% (SD = 0.2), respectively.

The findings were categorized into four themes: (i) understanding that T2D can be prevented, (ii) lifestyle changes made, (iii) hurdles to overcome, and (iv) experiencing benefits leading to sustained change.

### 3.2. Understanding That T2D Can Be Prevented

All participants were familiar with the term ‘diabetes’ for T2D, but no one had heard about the term ‘prediabetes’, although some had heard about ‘borderline diabetes’. Almost all participants verbalized that they had been unaware that diabetes could be prevented. It was said that this was a key motivator. A few participants spontaneously mentioned that they were happy to have been detected with ‘prediabetes’, because they understood that they had a chance to prevent T2D. For some, this was said to change their views on lifestyle modifications, as they had previously thought that T2D was inevitable.

*……I thought the disease (diabetes) might be hereditary but the (DiPEP) educator said that with diet control, it can be prevented*. (44 years, Female, Dig1)

Learning in the DiPEP educational sessions about what can be performed to prevent T2D was said to be an eye-opener. The participants talked about how changes in diet, physical activity, and stress management could prevent T2D. One example was obtaining new knowledge about different types of exercises taught in the sessions; these types were referred to as contributing to blood glucose regulation.

*My husband advised me to start jogging…But, what I learned [from the DiPEP classes] was that only aerobic exercise was not enough…there are four types (of exercise) that I know now…I should be doing all of these four types of exercises.* (55 years, Female, Dig8)

### 3.3. Lifestyle Changes Made

When participants were asked about lifestyle changes they had made, the topic most frequently mentioned was the reduction in food intake. Some said they used to eat large portions of rice with little vegetables in a single meal or drink lots of tea with sugar. After learning about the consequences of such a diet in the educational sessions, they said that they were motivated to change their diet. It was also reported that during the follow-up meetings, they found support in other participants talking about their own attempts to change their way of eating.

*……I used to have this big (indicated by hand) heap of rice, carbohydrate, on my plate…I have reduced at least a third of my previous portion size and added salad and vegetables. And,…I have quit soda…* (64 years, Male, Phy8)

Physical activity was also an area where there was frequent mention of changes made. This included both increasing the level of physical activity and adding different types of exercises. The reasons given were that the lessons in the sessions changed their perception of physical activity and exercises as important to prevent diabetes. One example was doing different types of exercise taught in the session, such as flexibility exercise, resistance exercise, and balance exercise, along with going for a morning walk every day as a part of moderate aerobic exercise. It was reported that they learned that instead of heavy exercises, they could do simple exercises at home to prevent diabetes.

*I have reduced my use of vehicles now. I use them only when it is absolutely necessary. Walking has become my habit now…I also do other light exercises. But I do not go beyond what my body can endure…* (47 years, Male, Dig10)

A few participants mentioned that before the DiPEP intervention, they had a regular habit of doing activities such as meditation and yoga to keep their minds at peace. They said that they would continue to do so as they had learned at DiPEP educational sessions that this was a good practice. For most of the participants, stress management was not an area where they said they had made changes; however, some said that they were surprised to learn about the link between stress and high glucose level and were motivated to implement some of the activities that were taught in the DiPEP educational session such as making a list of work for the next day before they went to bed to sleep.

*……I used to say ‘yes’ all the time and take on a lot of work…Now, I realised that I should say ‘no’ to the excessive work as [the DiPEP educator] taught me and I have already implemented it.* (44 years, Female, Dig1)

### 3.4. Hurdles to Overcome

Different participants stated different reasons for not being able to do what they were taught in the DiPEP educational sessions. Examples given were that they were too occupied with their household work and consequently were neither able to consider their diet pattern nor physical activity. One male participant stated that he did not have control over cooking food at home, so he had to eat whatever he was offered. Some reported that they were discouraged from doing exercises due to the cold weather (winter season).

Others had started to make changes but met different types of challenges, which made them return to their previous lifestyle. One participant talked about how he felt weak after reducing his daily food intake and doing vigorous exercise. In addition, he said that his mother was concerned for his health and therefore insisted that he increase his food intake again. He recommended that knowledge should be provided to the family in addition to the individual.

*When I reduced the amount of food, my mother commented ‘Oh, why are you eating so little? You will feel dizzy and become weak’*. (37 years, Male, Phy2)

Some experienced peer pressure and social cues during gatherings and parties, making it difficult to stick to the diet recommendations in such situations. Others talked about the struggle they had with themselves.

*…when I get sweets, I think for some time…whether I should eat this or not…But, I still end up eating some.* (44 years, Male, Phy7)

When we asked specifically about the effects of COVID-19 on their lifestyle changes, some participants did report that they increased their food intake, and some said they stopped doing exercises just to make sure they would not become weak during the times of the pandemic crisis. Two participants expressed that mobility restrictions hindered them from doing their regular exercises.

*I have an old house and it is not possible to exercise inside. I used to go outside…Due to lockdown, I cannot go outside…I have completely stopped exercising…* (42 years, Male, Phy3)

### 3.5. Experiencing Benefits Leading to Sustained Change

It was readily seen in the interviews that participants who experienced some positive outcomes from the changes they had made due to what they learned in the DiPEP lessons were motivated to continue. A frequently mentioned example was becoming aware of being overweight and starting to monitor it regularly on their own. When they noticed the reduction in weight, reduction in abdominal girth, or blood sugar down to normal level, they were even more motivated to continue the change.

*After reducing (amount) of rice and increasing vegetables and salad, I noticed reduction of my belly, which made me feel better. I think this is the result of choosing healthier food.* (52 years, Female, Dig6)

Finding that the exercises taught in DiPEP educational sessions were easy, safe, and efficient to do at home was another example of what they said motivated them to continue with the change. One of the benefits reported was having more energy with increased physical activity.

*…exercise has given me lots of benefits. I have an appetite and can sleep soundly.* (49 years, Male, Dig7)

This experience was also reported for some of the stress management strategies.

*……I have been doing as [name of educator] taught us such as making a list of work that I need to do the next day…and accordingly it is getting easier for me…*(44 years, Female, Dig1)

## 4. Discussion

The present study revealed that learning about the possibility of diabetes prevention at the prediabetic phase strongly motivated lifestyle modifications. It was reported that most changes were made in diet (reducing carbohydrate intake) and in physical activity (more active lifestyle), while fewer changes were made in stress management and in the handling of social situations. Lack of motivation, time, and family support meant that some did not change or reverted to their old habits. Experiencing benefits from lifestyle changes strengthened the motivation for maintaining the change.

‘Diabetes can be prevented’ was said to be an insightful message for the participants of the present study because they had thought that diabetes was inevitable due to heredity and thus not possible to prevent, similar to previous study findings [27,34,46]. Even though half of the participants in the present study had higher education, the information about diabetes prevention was still new to them. Evidence suggest that participants with a higher level of education had better participation in the lifestyle intervention [47] and had better results in terms of diabetes incidence reduction as a result of the intervention [48]. Studies also suggest that a diagnosis of prediabetes itself could motivate one to make lifestyle changes [30,34]. This is in congruence with the present study, where some participants expressed being happy to be detected with ‘prediabetes’. This new knowledge motivated participants of the present study to bring changes in their diet patterns and physical activity. The reported positive changes from the present study are in line with several other studies conducted worldwide [30,32,33,49,50].

Despite being aware of the possibility of preventing diabetes, people do not necessarily change their behavior [51]. Even if someone attempted to change behavior, lack of sustained and consistent effort at the maintenance stage of behavior change can lead to ‘relapse’ as described by the transtheoretical model [52]. Some of the participants of the present study reported having relapsed into their old habits due to different perceived social and psychological barriers, such as family and peers, similar to another study [30]. The barriers revealed by the participants of the present study, such as lack of personal motivation and the presence of external resistance, were similar to the challenges demonstrated in several past studies [35,53,54,55,56,57,58]. One male participant of the present study said that he ate what he was served, similar to the result of a study from the US [49], indicating that some male participants placed the responsibility of behavior change on the other members of the family. This might be due to the family cooking practices in Nepalese culture, where female members of the family cook and serve the food, and the other family members are expected to eat what is served. Traditional diet practices can also be a barrier, such as in Nepal, where it is common to start feeding rice to infants every day from 5–6 months of age and eating big meals with rice twice a day. On the other hand, changes in food culture, for instance, change from traditional food to ultra-processed food, may be the cause of obesity leading to prediabetes among genetically vulnerable individuals in the population [59,60]. In addition, participants neither knew about the adverse effect of obesity in developing prediabetes nor had any idea about required diet recommendations for good health and/or diabetes prevention [61,62].

Some of the participants of the present study were aware of the importance of exercise for good health, but not particularly for diabetes management or prevention. They also did not know about the minimum requirements of physical activity for good health [63]. Several male and female participants reported that morning walk was popular even before the DiPEP intervention. This was in contrast with studies in Bangladesh, where walking was perceived as an embarrassment and invited social criticism [33], and in Cameroon, where the morning walk was taken as a sign of poverty [46]. The intervention of the present study encouraged participants to do different types of simple exercises at home along with their morning walks. The introduction of simple exercises might have motivated participants to implement them. This is an example of the benefit of developing a curriculum addressing the local and cultural context.

Some participants in the present study reported having benefited from the intervention. This motivated them to maintain the changes they had made. This supports the notion that positive health outcomes and feedback encourage individuals to change their behavior [54,64,65] and sustain their changed behavior. It was also reported that some participants did not notice the immediate benefits of the intervention, such as a change in weight, despite reducing the amount of food. This means that some people with obesity might need long-term support to achieve a change.

### 4.1. Strengths and Limitations

There are several limitations to this study that should be considered while interpreting the results. This study was limited to relatively well-educated participants with prediabetes who were from 37–64 years old from one urban settlement in a developing country who participated in DiPEP intervention; therefore, the results cannot be generalized to other settings. Although the Nepali language was used in the interview, some participants could not express their thoughts well in this language as they normally used a different local language. This study did not explore the perception of participants who did not attend the educational sessions. The data were collected during the intervention period for participants who participated digitally and at the end of the intervention for participants who participated physically. Participants might not have talked about much of their negative experiences as the interviews were conducted by the main educator (PS). This might also have created confirmation bias. However, the interviewer (PS) made sure that the interviewing ambiance was neutral, and the participants also shared negative experiences. Before the interviews were concluded, participants were given the opportunity to say anything they wanted to say that had not been brought up during the interview. Furthermore, other authors took part in the analysis and found that although there were some examples of confirmation bias, this was not prevalent.

Strengths of this study include the use of semi-structured interview guides, which allowed for iterative and flexible probing of the questions to explore participants’ experiences and understanding. All interviews were individual, conducted in the Nepali language, and transcribed anonymously by native Nepali speakers to minimize loss of meaning that could have occurred during translation. The conduction of the interviews by the same person (PS) responsible for the educational sessions ensured prolonged engagement with the participants from the start of the intervention. The involvement of several researchers in the analysis of the data helped reduce subjectivity and increase trustworthiness [66].

### 4.2. Implication for Practice and Research

The findings of the present study indicate that individuals with prediabetes should be made aware of their current status and risks in the future by regular community-based screening in the context of low-resource settings such as Nepal. Awareness programs on diabetes prevention and its strategies in the community could be an aid to motivate individuals to prevent diabetes. The present study also demonstrated some of the aspects of lifestyle changes that people might experience during or after the intervention. Some experiences included positive changes, while other experiences included difficulty in implementing or sustaining those changes, as indicated by the transtheoretical model [52]. The findings can be used in developing new interventions in similar settings in low-resource countries incorporating the benefits and the hurdles mentioned by the participants, for instance, by emphasizing simple strategies of lifestyle changes such as adjusting the size of portions of a meal without introducing new foods, doing easy and safe exercises, etc. Future interventions could also be designed that include family members to ensure family support for lifestyle changes. Whether the changes reported and perceived barriers are still present in the long term is worthy of future investigation.

## 5. Conclusions

This qualitative study sheds light on the experiences of individuals with prediabetes who underwent community-based diabetes prevention intervention, which is one of the new areas of investigation for a low-resource country such as Nepal. The present study reveals the importance of the detection of prediabetes status among individuals. It also shows that understanding that diabetes can be prevented can be a key motivator for implementing lifestyle changes among persons with prediabetes in low-resource countries. It also highlights that consistent effort is a must to maintain the lifestyle changes made. The benefits and hurdles experienced by the participants of the present study can be taken into consideration while designing lifestyle intervention programs in similar settings.

## Figures and Tables

**Table 1 ijerph-20-05054-t001:** Sociodemographic characteristics of participants (n = 20).

Characteristics	Number
**Age**	
Range (years)	37–64
Mean (SD)	51 (9)
**Gender**	
Male	10
Female	10
**Occupation**	
Self-employed	10
Service	3
Home-maker	3
Other	4
**Education**	
Masters	3
Bachelors	2
Higher secondary	5
Secondary school	9
Primary school	1
**Mode of delivery of educational sessions**	
Physical session	10
Digital session	10
**Number of educational sessions attended**	
2 sessions	4
3 sessions	4
4 sessions	12

Table 1 presents the sociodemographic characteristics of the participants of the present qualitative study.

**Table 2 ijerph-20-05054-t002:** Baseline lifestyle characteristics, anthropometric measurements, and clinical characteristics of the participants (n = 20).

Variables	Number (%)
**Lifestyle characteristics**	
Smoking	
Non-smoker	19 (95)
Current smokers	1 (5)
Alcohol intake	
Never consumed alcohol	10 (50)
Current drinker	9 (45)
Former drinker	1 (5)
Physical activity (METs) (minutes per week)	
<600 METs/min	7 (35)
≥600 METs/min	13 (65)
Food intake (gm/day) Mean (SD)	
Total grain	637 (143)
Total fruits and vegetables	264 (106)
**Anthropometric measurements**	
Central obesity ^α^	
Yes	20 (100)
No	0 (0)
**Clinical characteristics Mean (SD)**	
RBS (mg/dL)	167.6 (33.9)
HbA1c (%)	5.9 (0.2)

^α^ as per WHO guidelines for Central Obesity for Asian population based on waist circumference (Female > 80 cm, Male > 90 cm). METs: Metabolic equivalent; gm/day: gram per day; RBS: Random Blood Sugar; HbA1c: Glycated hemoglobin. Table 2 shows baseline lifestyle characteristics, anthropometric measurements, and clinical characteristics of 20 participants who participated in the present qualitative study.

## Data Availability

The datasets used and/or analyzed during the present study are available from the corresponding author upon reasonable request.

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
