# Peer review of "How Did People with Prediabetes Who Attended the Diabetes Prevention Education Program (DiPEP) Experience Making Lifestyle Changes? A Qualitative Study in Nepal"

_ijerph, 2023, doi:10.3390/ijerph20065054_

Round 1

Reviewer 1 Report

Thank you for embarking in this kind of study. We need such studies to address the issue of diabetes. 

Reviewer 2 Report

Dear editor and authors, thank you for the opportunity to review this manuscript. The study is presented clearly overall, please see some more detailed comments on each section below. Please be aware that my decision cannot be understood as "minor" or "major" revision, but just as "revision necessary". 

General comment

·       The manuscript would benefit from a proof-reading by a native speaker, although it is generally well written; you often tend to forget a “the” like in “Diabetes can be prevented through lifestyle modification in prediabetic phase” > at least for introduction and discussion?!

Title

·       I think it should say “Experience with”, not “of” ?!

·       And there seems to be a “the” missing in “attending the diabetes prevention education program…”

·       I would change the title to: “How people with prediabetes who attend the Diabetes Prevention Education Program (DiPEP) experience lifestyle changes: a qualitative study in Nepal”

Introduction

·       There should be some more context about DiPEP, is it a particular intervention? Are there others like this one? Which institution is running it?

·       Is there previous qualitative studies that investigated prediabetes education perspectives?

·       The rationale for why your study is necessary should be stronger, you almost only mention that “more studies are needed for low resource countries”, but one could always argue that more studies for XY are needed. Can you give content-wise reasons for why a qualitative investigation here is relevant?

Methods

·       Please include the COREQ Checklist (Tong et al) to your methods section (can be in annex) and go through all the necessary items and add information for those that you haven’t considered yet in the methods

·       How did you compare the coding of the material among different researchers, i.e. have you come up with differences in coding among the individual researchers

·       The codebook should be accessible to readers, at least as an annex, though a short summary table in the methods or results would be preferable

·       Were n=20 participants enough to answer your RQ? If no more participants were willing, then please discuss if this was a limitation (or not) in the discussion/limitations

Results

·       Your sample seemed to be relatively well educated, right? Implications from this?

Discussion

·       Have you considered your results in terms of what people knew/learnt/do to prevent T2D and what medical/scientific evidence recommendations say about what should be done/what is effective?

·       I was a bit worried about the confirmation bias issue, are you sure that participants had a chance / felt able to express negative experiences? How did you achieve this from the interview guide/questions?

·       The implications for practice and conclusion could be more concrete: what are the most important aspects to include in interventions and education programmes more generally about what is necessary to increase awareness about the options for preventing T2D (of course, based on your results only, not what is known from other investigations)?

·       In the conclusion, or at the beginning of the discussion, please add a statement about what your study found that wasn’t known yet from other studies on the same issue

Reviewer 3 Report

The data presentation needs to be better organized

1) patients' clinical criteria are missing (e.g. blood glucose, hbA1c, smoking, excess weight, or BMI)

2) how did you diagnose prediabetes, did you perform any screening tests?

3) How had the subjects enrolled to participate in screening campaigns if they knew little or nothing about diabetes? Were screening on general population?

3) Of 20 patients selected from 31, it is unclear whether only 31 had participated in the screening programs. If there are so few, why were 11 discarded?

3) you report the incidence of prediabetes in Nepal (2020-9.2%), but not that of type 2 diabetes, what impact can a questionnaire have on 20 subjects?

5) the paper contains data and generic introductions on diabetes in general, but there are few references to the real situation of dm2 in Nepal

Round 2

Reviewer 2 Report

Dear Editor, the authors of this manuscript seem to have provided a thorough revision of the points I mentioned earlier, thus I am happy to recommend it for publication. Kind regards!

Reviewer 3 Report

I have read the corrections and the added material.

Better than the first version